# Sovereign Bond Yield Differentials across Europe: A Structural Entropy Perspective

**DOI:** 10.3390/e25040630

**Published:** 2023-04-07

**Authors:** Thierry Warin, Aleksandar Stojkov

**Affiliations:** 1Department of International Business, HEC Montreal, Montréal, QC H3T 2A7, Canada; 2Department of Business Law and Economics, Iustinianus Primus Law Faculty, Ss. Cyril and Methodius University, 1000 Skopje, North Macedonia

**Keywords:** structural entropy, sovereign bonds, European capital markets

## Abstract

This study uses structural entropy as a valuable method for studying complex networks in a macro-finance context, such as the European government bond market. We make two contributions to the empirical literature on sovereign bond markets and entropy in complex networks. Firstly, our article contributes to the empirical literature on the disciplinary function of credit markets from an entropy perspective. In particular, we study bond yield differentials at an average daily frequency among EU countries’ 10-year Eurobonds issued between 1 January 1997, and 4 October 2022. Secondly, the article brings a methodological novelty by incorporating an entropy perspective to the study of government bond yield differentials and European capital market integration. Entropy-based methods hold strong potential to bring new sources of dynamism and valuable contributions to the areas of macroeconomics and finance.

## 1. Introduction

The European economy was slowing quite visibly by the end of 2022. The risks stemming from the “complex uncertainty” (the multi-layered crisis comprising an ongoing pandemic, accelerating food and energy prices, and a security crisis in Europe) were being aggressively incorporated into the daily yield rates of sovereign bonds issued by the governments of the European Union (EU). This period also witnessed the fastest interest rate hikes by the European Central Bank (ECB), the Federal Reserve System, the Bank of England, and other central banks across the EU. Financial markets have become very sensitive and extremely careful when reexamining the macroeconomic fundamentals of each EU member state. In a nutshell, financial investors were rapidly repricing the cost of capital, which promptly resulted in widening bond yield differentials on the European government bond market.

Sovereign bond yield rates at a longer-term horizon are considered as long-term interest rates, i.e., one of the convergence criteria for Economic and Monetary Union (EMU) membership, based on the Treaty of Maastricht (1992). Before joining the monetary union, the yield on the EMU candidate’s long-term sovereign bond should not be higher than two percentage points above the average yield rate in the three best-performing EMU member states. Government bonds are also used by the private sector as a presumably risk-free financial investment, as collateral, and as an important benchmark for pricing other fixed income securities. Their yield rates, inter alia, reflect not only the financial markets’ perceptions of a country’s fundamentals and its debt-servicing capacity, but also global risk factors (e.g., global risk appetite and the global demand for credit). In general, an EU member state hit by an idiosyncratic shock would experience an increased sovereign bond yield rate relative to that of a 10-year German government bond (Bund), considered a high-quality financial instrument issued by a reliable borrower. Adverse shocks and prolonged crises, such as the multidimensional crisis, tend to increase sovereign bond spreads over the corresponding German bonds. We use sovereign bond yields because they include a risk premium and are a proxy for each country’s systematic risk. We could consider just the yield and look at the differences in yields across countries and through time, but we want to capture some dynamics in the yields. For that, we will compute the log differences of the yields. It will thus provide us with a structural entropy-based analysis of the changes in yields.

We make two contributions to the empirical literature on sovereign bond markets and entropy in complex networks. Firstly, our article contributes to the empirical literature on the disciplinary function of credit markets from an entropy perspective. In particular, we study bond yield differentials at an average daily frequency among EU countries’ 10-year Eurobonds issued between 1 January 1997, and 4 October 2022, by using data from *Bloomberg*. This period covers the initial enthusiasm with the introduction of the single European currency (1999), the 2008 Global Financial Crisis, the 2009–2012 European sovereign debt crisis, and the multidimensional crisis from 2020 until the present day. It also captures the short period before the introduction of the single European currency, as well as the evolving credibility of the ECB. Our main hypothesis is that, during crises, the yield rates of sovereign bonds in Europe are more loosely linked.

Secondly, this article brings a methodological novelty, by incorporating an entropy perspective to the study of government bond yield differentials and European capital markets’ integration. There are numerous definitions and formulations of entropy in physics. In general, entropy is used to measure information, surprise, or uncertainty regarding the potential outcomes of experiments. In particular, Shannon’s entropy [1] is one of the most commonly used methods in statistics and machine learning. In light of our objectives, we introduce structural entropy in complex networks. Almog and Shmueli [2] introduce structural entropy and apply it to the monitoring of a correlation-based network, over time, in the context of financial markets. We will use the same methodological approach to investigate country risk premia and European bond market integration.

The structure of the article is as follows. In Section 2, we will provide a critical review of empirical literature on sovereign bond markets from the standpoint of European convergence or divergence, as well as applications of entropy-based methods and complex networks in macro-finance. Section 3 elaborates on the data and the empirical strategy. Section 4 provides conclusions and policy recommendations.

## 2. Materials and Methods

### 2.1. Materials

Given our central goal, we construct a link between two strands of empirical literature: one on sovereign bond markets and EU integration, and the other on entropy-based methods in macro-finance contexts. In particular, we focus on the structural entropy in complex networks, which holds the potential to bring significant value added.

#### 2.1.1. Studies of Sovereign Bond Markets in Europe

Some of the earliest studies of European bond market integration were conducted by [3,4,5,6]. These studies focused on the degree of convergence (divergence) of market yields on the European government bond market in times of common or idiosyncratic shocks, as well as on economic policy changes. Their methodological approach was based on a correlation analysis over time of homogeneous financing instruments, such as the 10-year sovereign bond.

One strand of the empirical literature on sovereign bond markets in Europe is focused on the degree of financial integration. For example, [7,8] exploring the government bond yield rates as one of the price-based measures of European financial integration and monetary convergence, respectively. Kim et al. [9] examined the European bond market integration using daily bond yield rates during the 1998–2003 period. From a methodological perspective, they utilized dynamic cointegration, Haldane and Hall’s Kalman filtering method, and a bivariate version of [10] Exponential Generalized Autoregressive Conditional Heteroskedasticity (EGARCH) framework. They concluded that convergence appears to be slow, and that the pre-accession measures to achieve economic convergence are insufficient to generate rapid bond market integration. Holtemöller [11] investigated ex post deviations of sovereign bond yields from uncovered interest rate parity, using monthly data on interest rates and exchange rates of potential new EMU members from January 1994 to October 2004. His methodology for studying bond yield rate co-movement is based on backward recursive statistical tests and error correction models. Based on the variability of interest rate spreads, he classified EU accession countries into three groups: those with a low, medium, and high degree of monetary integration.

Another strand of the relevant body of empirical work explores the driving forces behind sovereign bond yields, or differentials. Gabrisch and Orlowski [12] empirically tested the volatility of the 10-year sovereign bond yields of the EU accession countries (the EU enlargement in May 2004) in relation to the euro area yields during the 2 January 2001–22 January 2009, period. By applying the GARCH model, they identified strong convergence between bond yields for most countries in Central and Eastern Europe and the euro area members. The divergence of bond yields in Hungary was found to be a combination of the country’s deteriorating macroeconomic fundamentals, and the contagion from the Global Financial Crisis. Ehrmann et al. [13] documented a substantial increase in the anchoring of long-term inflation expectations since EMU, particularly for Italy and Spain. They examined the yields across countries unconditionally, and then looked at the conditional correlations, using major macroeconomic data releases as the conditioning variables. Boffelli and Urga [14] examined the economically and statistically significant impact of macroeconomic announcements on government bond auctions in five EU member states, with respect to Germany. Afonso [15] explored the nexus between growth forecasts and fiscal positions, on one hand, and the government bond yield rates, on the other hand.

A third strand of the empirical literature is focused on spillover effects and shock propagation across the European government bond market. Balli [16] investigated the various levels of response of each euro market to global shocks, and discovers that euro bond markets are not fully integrated with one another. Antonakakis and Vergos [17] highlight that sovereign bond yield spread spillovers between euro area countries are quite common. Their main conclusions are that (a) present shocks tend to increase future bond yields; (b) shocks from the peripheral European economies have, on average, a more destabilizing influence on other countries than shocks coming from the euro area core; and (c) joint shocks produce decoupling between peripheral and core countries. More recently [18], used fractional cointegration analysis to explore the relationship between German and Spanish government bond yields with various maturities in the 2011–2006 period. They also conclude that the differential—between the yields of the two countries in all maturity brackets—does not react to the level of interest rates in the currency union. Spillover effects of different maturities of government bonds was also documented by [19]. From a methodological perspective, we also outline the system-risk investigation of spillovers on the Chinese bond market [20].

#### 2.1.2. Studies of Structural Entropy in Complex Networks

Before going further, let us step back a bit and reflect on the methodological aspect. Our article does not claim to be the one that starts the conversation on entropy in finance and economics. At best, we provide a first contribution to the applied context of European convergence. However, we are convinced that our article is relevant, in that it contributes to the multidisciplinary conversation intersecting physics and finance/economics (e.g., [21,22]). This conversation is more about fundamental research, while our article is an empirical contribution. Mirowski [23] uses a personal anecdote to illustrate his own interest in the subject matter. He recalls a comment made during an economics seminar at Stanford University in the late 1970s or early 1980s. The speaker casually made an observation suggesting that “value had to be conserved in his model if some mathematical assumption in the model were to hold; the tone of his voice suggested that no one in his or her right mind would find that a problem” [23] (p. 2). The issue of conserving value can serve as a basis for exploring models from physics, specifically those related to energy conservation, such as heat and thermodynamics. According to [23] (p. 12), “my first task is to convince the reader otherwise: that there is no way of understanding economics and social theory in the twentieth century without first understanding “energy” in some detail; and further, that knowledge of energy will still prove inadequate unless it is embedded within a familiarity with the historical development of the energy concept. It will subsequently prove useful to see that the energy concept traced a trajectory from being thought of as a substance, to being regarded as a mathematical artifact of a field, to being just another way to express symmetry. Similarly, the timing of various changes in the way scientists thought about force and the conservation of energy will later be critical in explaining certain developments in economic thought with respect to the theory of value”.

The notion is not novel; political economists and later, neoclassical economists, have developed a collection of theories that bear some resemblance to the advancement of mechanical physics. In his book *Le Monde*, Descartes elevated the following three principles to the status of natural laws: (1) the conservation of modes of bodies, including motion and rest, in the absence of perturbing factors; (2) the conservation of the total quantity of “force of motion” in collisions between bodies; and (3) the determination of this force of motion to act in a straight-line tangent to the path of the body [24] (p. 632). Afterward, Leibniz drew inspiration from Descartes’ work to construct his own philosophical system, where he utilized calculus to explain that system, and engaged in the first extensive discussion of physical conservation principles [23] (p. 18). Maupertuis, Euler, Lagrange, and later, Carnot, played crucial roles in establishing connections between mechanics and energy. However, the concept of entropy is comparatively more elusive than that of energy. As underscored by [25] (p. 113), “entropy is the physical measure of disorder. All energy conversion processes produce entropy. Entropy production is coupled to emissions of heat and particles”.

The penetration of entropy-based methods has brought new sources of dynamism and valuable contributions to the areas of macroeconomics and finance (e.g., [26,27,28]). Conventional econometric methods have proven insufficient in elucidating market turbulence and crises. In contrast, entropy-based methods provide a dynamic portrayal of financial markets as correlation-based networks. These approaches expand our comprehension of financial markets, by conceptualizing financial assets as nodes in a network, and their price fluctuations and interactions as edges in the network.

Recent empirical work used the concepts of “transfer entropy” (e.g., [29,30,31]), “permutation entropy” (e.g., [32,33]), “spectral entropy” (e.g., [34]), “moving average cluster entropy” (e.g., [35]), “dispersion entropy” (e.g., [36]), “natural time entropy” (e.g., [37,38,39]), etc. As an illustrative example, [31] employ the transfer entropy method to examine information flows between sovereign Credit Default Swaps and equity returns in 9 Central and Eastern European economies, utilizing daily data over a 2008–2018 period. Meanwhile, Kozak et al. [33] use symbolic representation to simplify financial data complexity, and use permutation entropy as a gauge of the information gain or loss for financial data.

One of the main advantages of the structural entropy method, is the facilitation of continuous monitoring of the level of structural diversity of a network over time. Our literature review is limited to the applications of structural diversity and structural entropy in complex networks, specifically in the context of financial markets. In our context, structural diversity refers to the number of connected components in the network, specifically sovereign bond issuers in Europe. Meanwhile, structural entropy is a novel measure, inspired by the Shannon Index, that gauges structural diversity by not only counting the number of node groups and their sizes, but also by continuously monitoring a correlation-based network. It is worth noting that an alternative method, beyond the scope of our paper, regards financial networks as dynamic asset trees and spanning trees (e.g., [40,41,42]).

Almog and Shmueli [2] develop a simplified version of the structural entropy method, where each node is assigned to a single community. They then compare the outcomes of this method to those of spectral entropy and Chaikin volatility, and find that structural entropy effectively anticipates a period of market volatility long before it is observed.

It is now an established tradition to treat financial markets as complex networks. For illustration only [43], analyzed a new dataset on the USD 1.2 trillion Federal Reserve’s emergency loans program to global financial institutions during 2008–2010, and identified systemically important institutions (nodes) in the network. Münnix et al. [44] analyzed financial data from the S&P 500 stocks in the 1992–2010 period. They propose a definition of “state” for a financial market, and use it to identify points of significant change in the correlation structure. By using this classification, they recognize transitions between different market states. More recently [45], investigated the evolution of short-term cross-correlation structure patterns during the 1985–2016 period, and conducted a comparative analysis of two stock markets: the S&P 500 (United States) and the Nikkei 225 (Japan). Their contribution is in identifying precursors to volatile periods, when designing an early warning system for financial markets. By analyzing the daily time series of the logarithm of stock prices, [46] concentrated on the hierarchical organization of equities traded on a financial market.

The rapidly growing body of research on financial markets as complex networks, suggests that the structural entropy method will be used more frequently in macro- and micro-financial research. To the best of our knowledge, this is the first article to use structural entropy in studying the complex network of the European sovereign debt market.

### 2.2. Model

The term “diversity” in the context of a network generally pertains to the level of dissimilarity among the various agents within the system, and is frequently interpreted as the number of connected components in the network. Almog and Shmueli [2] proposed a new measure, which they coined “structural entropy”, as a revised interpretation of “structural diversity”. The proposed measure is based on the more detailed network communities, rather than solely the network’s connected components. This measure accounts for both the quantity of communities and their respective sizes, resulting in a single representative value [2].

The application of structural entropy is warranted, due to the limitations of traditional econometric models that rely on the assumptions of rational agents and efficient markets. While these models can be effective during stable market conditions, they may not be sufficient during times of significant market disruption, particularly in terms of capturing intricate system interactions. Given these challenges, we deem it necessary to explore the use of symbolic methods, as outlined in [2], in order to effectively assess and navigate these complexities.

We make use of the concept of structural entropy, a measurement of how structurally diverse a network is. It originates from symbolic representations and network theory, and captures the degree of heterogeneity among nodes in a network.

The representation of sovereign bond yield rates as correlation-based networks, is a relatively recent development. In the context of European sovereign bond markets, the network’s nodes symbolize the daily yield rates of sovereign bonds, while its edges represent their interactions. For example, a sudden increase in the yield of a core euro area country’s sovereign bond (such as Germany) would likely trigger a surge of higher risk premia and bond yields for other euro area members. Firstly, the price of the 10-year German bund is used as a benchmark to price other sovereign bonds across Europe. Secondly, due to the high degree of international trade and financial integration, any potential issues that arise within the German economy are likely to impact other European economies and raise their government borrowing costs. These interactions are typically evaluated by analyzing the correlation strength of sovereign bond yield rates over time.

Structural entropy is a measure that calculates the level of structural variability within a network based on its community structure. This measure uses the concept of network node diversity, which refers to the degree of interconnectedness among nodes with similar roles or characteristics, compared to nodes with diverse roles or characteristics. The calculation of structural entropy is based on a broad framework for correlation-based networks, which involves several simple steps. The network’s community structure is used to compute structural entropy, which provides a more detailed breakdown of the network into sub-units than do basic linked components. Moreover, structural entropy considers both the number of communities and their sizes, thus providing a more comprehensive and meaningful assessment of the network’s structure, in a single value.

Entropy has diverse meanings and formulations, including measuring information, surprise, or uncertainty about the possible outcomes of experiments. The most commonly used formulation in statistics and machine learning is the Shannon entropy. In this context, the Shannon Index is generalized and adapted to evaluate the structural diversity of complex networks, as in [2]. (1) An *N***N* (*N* = number of time series in our panel data) symmetric matrix is produced by first computing the Pearson’s correlation coefficient for each series.
(1)$$\rho_{X,Y}=\frac{E\left[\left(X-\mu_{X}\right)\left(Y-\mu_{Y}\right)\right]}{\sigma_{X}\sigma_{Y}}$$

Let us now consider the same formulation as in [2] a network G with *N* nodes, and let A be the chosen community detection algorithm. An *N*-dimensional vector can be used to represent the division of nodes into communities, as generated by applying A to G, where the *i*-th component (*σ_i_*) represents the community to which node *i* belongs. The values range from 1 (community 1) to *M*, which represents all communities that were found. We determine the *M*-dimensional probability vector *P*, which depicts the proportionate size of the clusters in the network, given the partition.
(2)$$\;P\equiv \left[\frac{c_{1}}{N},\frac{c_{2}}{N},\ldots ,\frac{c_{M}}{N}\right]$$
where *c_i_* is the size of each community, and *P* denotes the likelihood of selecting a node at random from each community (note that $\sum_{i=1}^{M}P_{i}=1).$

The network’s edges are represented by an adjacency matrix that is constructed from the correlation matrix in a variety of methods. The most common approach is to utilize threshold criteria to determine which correlation matrix values will be network edges, and which values will not. The number of significant clusters meeting a specific threshold, based on the matrix’s structure, can be used to determine entropy. For example, if all variables have a meaningful correlation, there will be only one cluster, and there is very little correlation information in the multivariate time series. When all *N* time series’ are uncorrelated, the correlation information in the multivariate time series is maximized, resulting in *N* clusters.

After the clustering process, the labels (a vector of integers) are used to calculate the traditional Shannon entropy, with the cluster count frequencies being utilized to compute the entropy. The resulting value is known as the network’s structural entropy. It is worth emphasizing that a new time series of structural entropy values is generated for each relevant sub-period at every time step, enabling the computation of structural entropy.

We apply the probability vector to Shannon entropy:(3)$$\;S\equiv H\left(P\right)\equiv -\sum_{i=1}^{M}P_{i}log\left(P_{i}\right)$$
and the resulting number is known as network G’s “structural entropy”.

A sliding window technique that combines these activities, enables us to follow the time-dependent dynamics of the system of interest. A fresh, single time series of structural entropy values, is the result. A structure that is geographically and temporally complex, like our own dataset, can be thoroughly analysed by employing the structural entropy method.

As presented in Figure 1, a new single time series of structural entropy values is the result.

In essence, structural entropy is a modified version of the well-established Shannon index, with both serving as diversity indices. However, structural entropy is capable of extracting value from a structure that is spatially and temporally intricate, making it particularly useful in the context of complex systems such as financial markets.

## 3. Data and Empirical Strategy

We break down our empirical plan into numerous steps.

In order to conduct our research, we first create a dataset of daily 10-year sovereign bond yields. We collected data for 25 countries (*N* = 25), from 1 January 1997, to 2 October 2022, for a total of 8015 observations.

The countries are: Austria, Belgium, Bulgaria, Cyprus, Czechia, Denmark, Finland, France, Germany, Greece, Hungary, Ireland, Italy, Latvia, Lithuania, Malta, Netherlands, Poland, Portugal, Romania, Slovakia, Sweden, and Spain.

A time series of each country’s daily log-returns was constructed for the dataset. The sovereign bond yield value (log-returns) can be log-differentiated to provide stationary, normally distributed signals.
(4)$$s_{i}\left(t\right)\equiv ln\left(\frac{p_{i}\left(t\right)}{p_{i}\left(t-1\right)}\right)$$
where *p_i_*(*t*) is the daily bond yield per country. Then we calculated log-differenced yields’ correlations and distributions (Figure 2).

Step 2: Using standard measurements (such as the mean and standard deviations of the time series), we start looking into whether there is turbulence in the data. The matching correlation-based network was built using the sliding window method, and each sub-period was then evaluated (Figure 3). The selection of an appropriate time period for constructing the correlation matrix is a well-known concern in the field of correlation-based networks. There is a clear trade-off between long and short intervals. Long intervals reduce volatility and noise, but may not be stationary. Conversely, short intervals produce correlation matrices that are highly specific and exhibit significant fluctuations.

To address this issue, we chose quarterly data, to strike a balance between these competing factors. This way, if we observe significant fluctuations, they are more likely to be due to rare occurrences rather than the inherent volatility of the data.

The standard deviation is a commonly used indicator of volatility, serving as a statistical measure of the dispersion of quarterly returns on government bonds’ interest rates or alternatively, of the volatility. This metric is particularly intriguing, as it reveals the degree of risk or uncertainty associated with variations in sovereign bond yield rates’ magnitude. High volatility is correlated with a broad range of yield rates on sovereign bonds and market flow variations. As a result, the value of the flow of sovereign bond yield rates, is subject to rapid fluctuations.

When there is less volatility, yields on government bonds move less frequently and are more likely to remain stable. This relationship will enable us to evaluate how the dispersion of bond yields, acting as a proxy for the convergence status among European countries, is related to our study’s subject.

The sliding mean and sliding standard deviation started to fluctuate after the 2008 Global Financial Crisis, but the variance was very small between 2008 and 2014. Then, between 2014 and 2020, we see a significant amount of variation. Last but not least, the COVID-19 years exhibit the highest sliding mean and sliding standard deviation dispersions.

Step 3: Following this, as outlined in the model section, we calculate the structural entropy over a defined period of time. We utilize a community discovery technique, such as connected components, to the adjacency matrix. By employing structural entropy, which demonstrates the flexibility of the requisite correlations, we can transform a collection of time series, into a signal time series comprising entropy values. This adjustment is highly beneficial for tracking correlations and detecting outliers.

When we apply the entire process to the available data, it produces a series of outcomes, as shown in Figure 4.

The structural entropy is at its lowest possible value, which is zero, when every node in a network is a member of the same large community. When each node represents a different community, the maximum is attained. As we can observe in Figure 4, structural entropy is able to account for part of the instability that existed in the years following the 2008 Global Financial Crisis. Sovereign bond yield rates typically remain independent and cluster into discrete groups during times of highest structural entropy. We can see that some sovereign bond yield values belong to the same cluster and are strongly connected during the time of moderate entropy. When entropy is low, most sovereign bond yield rates belong to the same group and exhibit a wide range of correlations.

To us, it is fascinating to realize that we can track European convergence (and divergence) cycles through structural entropy measurement.

Investigating the change in the structural entropy index versus standard deviation could be conceptualized as a robustness question. That is exactly what we would like to do next (Figure 5).

The calculation of structural entropy’s edge cases can be examined, and it can be contrasted to volatility (see Figure 5). According to our analysis of this robustness component, structural entropy offers a more accurate depiction of convergence, compared to the standard deviation. The standard deviation is calculated in an incremental way, by incorporating the new value in each consecutive day of trading. For instance, between 2000 and 2004, we see an increase in volatility, and a decrease in structural entropy. And on the same graph, these phenomena may be seen quite a few times. The simple reason has to do with the fact that structural entropy captures the complexity of the system, while standard deviation is an examination with regard to the mean.

## 4. Discussion

Our study employs structural entropy as a valuable tool in studying macro-finance questions. This method can help identify patterns and correlations between different parts of the network, and can be used to measure the stability and robustness of the network. It can also be used to identify structural characteristics that are associated with network performance and resilience. Structural entropy is used as an alternative—yet innovative—method for analyzing market variations of European sovereign government bond yield rates.

The article contributes to the empirical literature on the disciplinary function of credit markets from a structural entropy perspective. In particular, we study bond yield differentials at an average daily frequency among EU countries’ 10-year Eurobonds traded between 1 January 1997, and 4 October 2022, by using data from *Bloomberg*. The European government bond yield rates fluctuate a lot: a simple change in the information causes an immediate change in the yield rate. The interconnected nature of sovereign bonds and spreads, calls for the adoption of a spatial and temporal dimension-capturing metric. In this way, structural entropy facilitates the formulation of statements and the execution of computations regarding uncertain situations.

The turmoil that existed in the years following the 2008 Global Financial Crisis can be well captured by structural entropy. Sovereign bond yield rates typically remain independent and cluster into distinct groups during times of maximum structural entropy. We also observe that some sovereign bond yield values belong to the same cluster and are strongly connected during times of moderate entropy. In times of low entropy, most sovereign bond yield rates belong to the same group and exhibit a wide range of correlations.

The representation of financial markets as networks offers substantial potential for a comprehensive investigation of financial markets, particularly the European sovereign bond markets. Numerous potential research avenues may be pursued in the future, such as examining the impact of euro area membership, the effects of joining the euro area, the factors that drive entropy, the internal structure of communities and the relationships between them, and so on. These investigations may yield a deeper understanding of financial markets and their dynamics, leading to more effective decision-making and policy implementation.

## Figures and Tables

**Figure 1 entropy-25-00630-f001:**
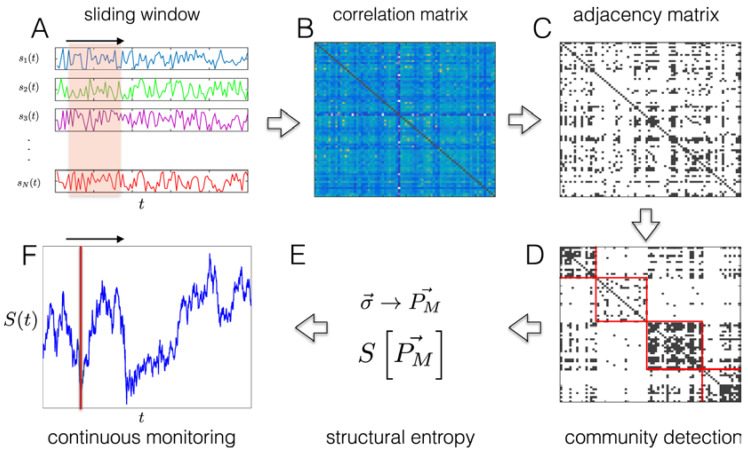
Monitoring structural entropy of correlation-based networks. Source: [2] (p. 4). Note: Illustration of the main steps in the procedure of monitoring structural entropy of correlation-based networks. (**A**) raw time series, using sliding window approach we continuously analyse sub-periods from the entire data; (**B**) generating Pearson correlation matrix (or different association matrix); (**C**) transforming the correlation matrix into an adjacency matrix; (**D**) resolving community structure of the network; (**E**) calculating the structural entropy for each specific sub-period; (**F**) continuous monitoring of the structural entropy and analysis.

**Figure 2 entropy-25-00630-f002:**
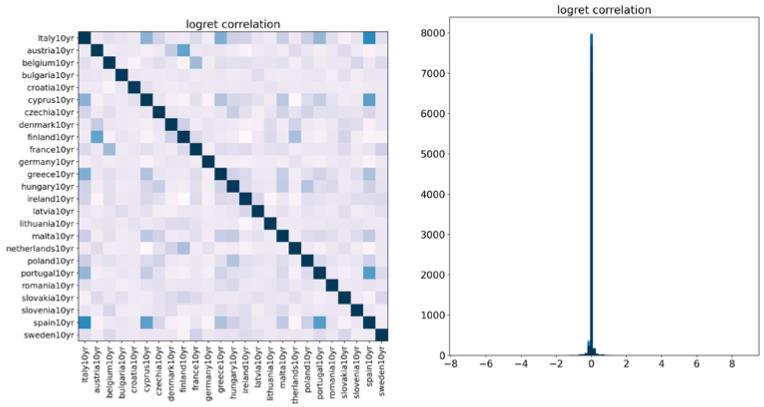
Log-differenced yields’ correlations and distributions.

**Figure 3 entropy-25-00630-f003:**
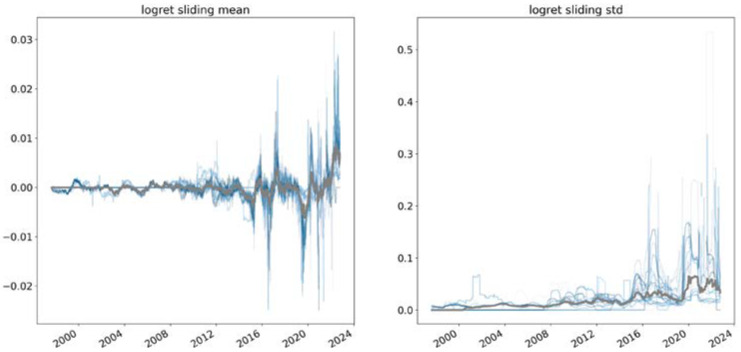
Sliding mean and sliding standard deviation of log returns. Note: “Logret” refers to the natural logarithm of the returns on sovereign bonds.

**Figure 4 entropy-25-00630-f004:**
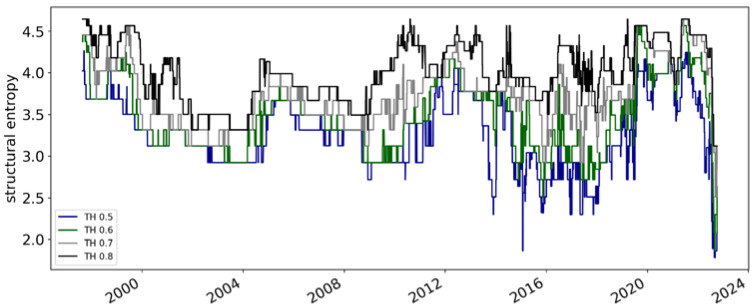
Structural entropy over time. Note: “TH” stands for different values of threshold criteria.

**Figure 5 entropy-25-00630-f005:**
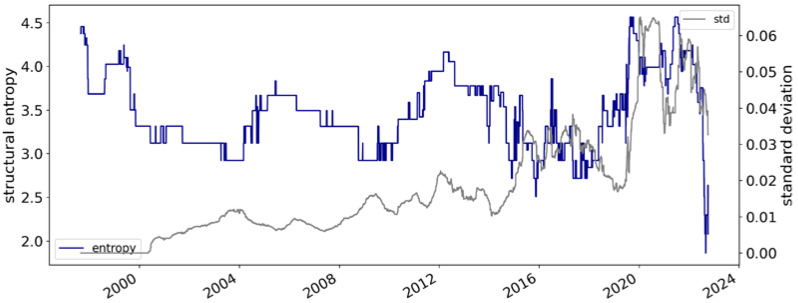
Structural entropy vs. standard deviation.

## Data Availability

Not applicable.

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
