# Peer review of "Sovereign Bond Yield Differentials across Europe: A Structural Entropy Perspective"

_entropy, 2023, doi:10.3390/e25040630_

Round 1

Reviewer 1 Report

The presented article makes two contributions to the empirical literature on sovereign bond markets and entropy in complex networks. Firstly, the authors contribute to the literature on the disciplinary function of credit markets by examining bond yield differentials among EU countries' ten-year Eurobonds issued between 1997 and 2022. The use of data from Bloomberg allows for a detailed investigation of significant events such as the global financial crisis and the European sovereign debt crisis, and the inclusion of data before and after the introduction of the single European currency is commendable.

Secondly, the article introduces a methodological novelty by incorporating an entropy perspective to the study of government bond yield differentials and European capital markets' integration. The authors explain the concept of entropy and how they apply it to their research, specifically introducing structural entropy in complex networks. The use of this approach allows for monitoring a correlation-based network over time in the context of financial markets, which is a valuable contribution to the field.

The article provides a comprehensive and innovative contribution to the empirical literature on sovereign bond markets and entropy in complex networks. The authors' use of a novel methodological approach and their analysis of bond yield differentials during significant economic events are commendable.

However, there is a remark to the presented research. Authors presented comprehensive literature overview on classical econometric approaches studied for European bond market and mention in a few sentences other entropy methods such as “transfer entropy” or “permutation entropy”, but they do not tell why structural entropy approach is more prominent comparing to classical ones. It is obvious for those who dealt with them but remains unclear for unexperienced readers. It would be also great if authors gave more references on those entropies, which were studied for different types of financial markets.

Overall, the article is a valuable resource for scholars and practitioners interested in the disciplinary function of credit markets and European capital markets' integration.

Author Response

Thanks for your comments, and please find our responses here.

Reviewer 2 Report

In this manuscript (ms), the authors present a study of the sovereign bond yield differentials across Europe. They employ the method of structural entropy within the adjacency matrix and in particular within the different communities identified in this adjacency matrix (see Figure 1). The results obtained are interesting and original as concern the European Union application. The research subject in very interesting and the ms falls within the scope of both the special issue and the journal of Entropy.

The presentation of the ms, however, needs significant improvement in order to be readable from the audience of Entropy, the majority of which are physicists and not economists.

Specifically, in a revised version the authors should consider the following point for improvement:

1)ll.58-59 “introduction of the single European currency (2009–10)” the mentioned years are probably wrong.

2)ll.79-80 “2. Materials and Methods” should appear as a Section title.

3)The authors of a citation is better to be mentioned when a sentence starts with a citation number. For example, in l. 97 “Kim et al. [9] examine” instead of simply “[9] examine”. The same applies to the citations starting sentences in ll. 103, 111, 118, 121, 123,127, 129, 148, 193, 200.

4)l.171  “p.632.”

5)In ll.181-183, the authors present various entropic measures that might find applications in economics. For the readers’ better information, the authors might also mention, in the same list, natural time entropy [Varotsos et al. "Entropy in the natural time domain", Physical Review E 70 (2004) 011106, https://doi.org/10.1103/PhysRevE.70.011106 ; Mintzelas and Kiriakopoulos, “Natural time analysis in financial markets”, Algorithmic Finance 5 (2016) 37–46, https://doi.org/10.3233/AF-160057,  Sarlis “Entropy in Natural Time and the Associated Complexity Measures”, Entropy 19 (2017) 177, https://doi.org/10.3390/e19040177 ]

6)l.219 “the network. proposed” please include who proposed.

7)l.228, “[2]’s” should better be written as “Almog and Shmueli’s [2]”

8)ll.233-241. This paragraph needs to be rewritten because it is very difficult for a physicist or mathematician to understand what the authors suggest. A reference to the time-series shown in panel A of Figure 1 would be very useful to simplify and clarify what the authors do.

9)l. 257, “N” is not a time series but rather the number of time-series used (N=25) if I understand correctly.

10)l. 263, “where the i-th component I represents the community to which node I belongs” is incomprehensible.

11)l.268 “c_i”

12)l.291 “can yield value through the use of structural entropy.” I cannot follow what the authors want to say please reword.

13)l.309, “for N=25 countries”, if understand correctly N equals the number of countries and it is better for the readers’ understanding to be defined somewhere in the ms.

14)l.319, “p_i(t)”

15)ll.324-333. Does this paragraph discuss an averaging period used? Please explain its relation with the results obtained otherwise it looks like a general paragraph that might exist in the discussion section.

16)I could not find reference to Figures 2 and 3 in the text.

17)l.348, Text is missing before this line.

18)What is TH mentioned in the legend of Figure 4? Please explain.

19)ll.372-377, this is a repetition.

20)l.385, please explain  to which standard deviation the authors refer to.

21)Why two figures (Fig. 5 and 6) are needed while all quantities appear in Figure 6?

In summary, I would be glad to suggest publication of a revised version of the ms in which the authors have taken in to account the points mentioned above.

Author Response

Thanks for your comments. Please find our responses here. We hope you find them useful. We do believe the paper is really better now. Thank you.

Round 2

Reviewer 1 Report

Since almost all comments and recommendations have been taken into account by the authors, I believe that the article can be published in its present form.

Reviewer 2 Report

The authors have appropriately addressed all the points of my previous review in this revised version. I am glad to suggest publication in the present form.